# Assessing Patient Satisfaction and the Need for Collaborative Treatment with Korean and Western Medicine

**DOI:** 10.3390/healthcare12181901

**Published:** 2024-09-23

**Authors:** Soyong Park, Yoonju Lee, Linae Kim, Shiva Raj Acharya, NamKwen Kim

**Affiliations:** 1College of Korean Medicine, Pusan National University, Yangsan 50612, Republic of Korea; oobsy1@pusan.ac.kr (S.P.); y.juleeee@gmail.com (Y.L.); 2Research Institute for Korean Medicine, Pusan National University, Yangsan 50612, Republic of Korea; klinae@pusan.ac.kr (L.K.); sameeracharya39@gmail.com (S.R.A.)

**Keywords:** collaborative treatment, patient satisfaction, Korean medicine, Western medicine

## Abstract

Background: The collaborative treatment of Korean medicine (KM) and Western medicine (WM) in Korea has gained prominence since its initiation. However, comprehensive evaluations of patient satisfaction and care effectiveness remain limited. Thus, this study aimed to evaluate patient satisfaction and the need for collaborative KM-WM treatment in the fourth phase of the national pilot project. Methods: A multicenter survey was conducted among 321 patients from 15 institutions participating in the fourth phase of the collaborative KM-WM pilot project, spanning from 1 August 2023 to 31 October 2023. Patient satisfaction and needs were assessed using a validated, semi-structured questionnaire with a 5-point Likert scale. Descriptive statistics and hierarchical multiple regression were used in the analysis. Results: The overall satisfaction with collaborative KM-WM treatment was notably high (91.25%). Among the participants, 91.58% indicated the necessity of the collaborative KM-WM pilot project, whereas 90.66% pointed out the need to incorporate inpatient services into collaborative care. Expansion of the pilot project to additional institutions and primary healthcare settings was substantially demanded (85.36% and 80.06%, respectively). Treatment effects (*β*, 0.344; 95% CI: 0.237–0.451), appropriate treatment time (*β*, 0.140; 95% CI: 0.051–0.229), medical procedural efficiency (*β*, 0.227; 95% CI: 0.126–0.328), and promotional activities (*β*, 0.175; 95% CI: 0.101–0.250) significantly contributed to overall patient satisfaction (each, *p* < 0.05). Conclusions: The fourth phase of the KM-WM project reflects high patient satisfaction and a substantial need for collaborative treatment. Further research should include longitudinal studies and employ mixed-methods approaches to better understand, evaluate, and improve collaborative KM-WM treatment.

## 1. Introduction 

The concept of collaborative Korean medicine (KM) and Western medicine (WM) treatment programs, hereafter referred to as collaborative treatment, involves the joint practice of WM and KM practitioners [1,2]. In 2009, the Korean government permitted mutual employment among medical, dental, and KM practitioners. This change also enabled the establishment of specialized departments in both KM and WM hospitals, as outlined in Article 43 of the Medical Service Act, to explore and promote new medical services. The amendment to this act took effect in January 2010 [3]. However, owing to this amendment by the Ministry of Health and Welfare, patients who received both KM-WM treatments for the same disease on the same day were not covered by health insurance for the cost of the subsequent treatment, leading to an increased financial burden on patients [4,5]. However, by 2015, 68.5% of KM hospitals and 4.6% of WM hospitals had established collaborative treatment practices [4].

The Korean government launched a pilot project for collaborative KM-WM treatment in July 2016 to address efficacy, safety, and cost-effectiveness and to enhance insurance coverage [6]. In the first phase, health insurance coverage was applied to both KM-WM treatment costs when collaborative treatment was provided for the same disease on the same day [6,7]. The second phase implemented an additional collaboration fee if specific diseases or conditions were treated with collaborative care. The third phase, which commenced in October 2019, maintained the basic structure of the second phase and introduced the quality-based differential treatment fee model for participating institutions. By the fourth phase in 2022, the project expanded to 75 institutions to systematize collaborative treatment further and strengthen research on its effectiveness [4].

Patient satisfaction is a crucial determinant of healthcare quality, equity, service continuation, adherence to treatment, and overall health outcomes [6,8]. Prior studies have shown high levels of satisfaction among patients who have experienced collaborative treatment [7,8,9]. Specifically, in the second phase of the pilot project, the overall satisfaction rate for treatment effectiveness was 89.56% [8]. Furthermore, a study revealed that participants expressed a more positive attitude toward the necessity and efficacy of collaborative treatment [10]. Interestingly, research among healthcare providers revealed that nearly 80% of the nurses supported the demand for collaborative treatment [11], whereas 62.5% of doctors were satisfied with it [12]. In contrast, other studies have indicated that over 50% of respondents hold negative perceptions toward collaborative treatment, citing low efficacy and cost-effectiveness [9,13]. Assessing the need and willingness to pay for collaborative treatment is critical to understanding the personal benefits of healthcare services perceived by patients [7,9,14]. Despite the growing collaborative treatment practices in Korea, the financial burden on patients for subsequent treatments still presents a challenge. This limitation in coverage may constrain their access to and the affordability of the full spectrum of collaborative treatment [1,9]. Thus, understanding patient perspectives on service quality can identify gaps in care and inform decision-making to improve and expand collaborative treatment [2,14,15].

Although there have been some studies on patient satisfaction with collaborative treatment [16,17,18,19], there remains a need for a comprehensive and systematic evaluation of quality, satisfaction, and need across different phases of collaborative treatment. Some studies have noted that negative perceptions of collaborative treatment are largely driven by misunderstandings, lack of trust, and significant gaps in research evidence [9,11]. In addition, substantial unmet practical and technical onsite demands persist for collaborative treatment. Therefore, this study aims to fill this gap by providing a multicenter assessment of patient satisfaction and the need for collaborative treatment in the fourth phase of the pilot project. By focusing on the quality of service and patient experiences, this research offers valuable insights into how collaborative treatments meet patient expectations and identifies areas for improvement. These findings are essential for guiding future policy decisions, addressing patient health needs, and further enhancing overall treatment outcomes. The objective of the study was to assess patient satisfaction and the need for collaborative treatment based on the quality of service provided by the enrolled institutions.

## 2. Methods

### 2.1. Study Design and Settings

A multi-institutional, cross-sectional survey was conducted across 15 institutions participating in the fourth phase of the collaborative KM-WM pilot project (Appendix A). These institutions were selected on the basis of their quality grade, collaborative treatment infrastructure, and research capabilities. Patients aged 19 years and above who received collaborative treatment at these institutions during the project period and incurred relevant treatment fees were included in the study. Patients who had difficulty reading and writing or who had mental or physical impairments preventing them from understanding and responding to the survey questionnaire were excluded. The study was conducted from 1 August 2023 to 31 October 2023.

### 2.2. Study Variables 

The outcome variables of the study were patients’ satisfaction with and need for collaborative KM-WM treatment. Patient satisfaction was evaluated using an 8-item questionnaire with a 5-point Likert scale, ranging from 1 (strongly disagree) to 5 (strongly agree). Patients’ need for collaborative treatment was measured using a 7-item questionnaire with a 5-point Likert scale, ranging from 1 (strongly disagree) to 5 (strongly agree). Both the patient satisfaction and the need scale showed good internal consistency, with Cronbach’s alpha values of 0.86 and 0.82, respectively. The independent variables included gender (male or female), age, purpose of visit (acute or chronic treatment), duration of visit to institution for collaborative treatment (in months: 1; 1–6; 6–12; >12), route of treatment (KM to WM or WM to KM), motivation for using collaborative treatment, and trust in KM (very high; high; moderate; low; very low). 

### 2.3. Data Collection 

Data were collected from participants using a semi-structured questionnaire. The questionnaire included sections on informed consent, sociodemographic information, and items related to satisfaction and the need for collaborative treatment. This questionnaire was developed on the basis of existing literature [7,9,15,18], the clinical experience of researchers, and discussions with experts from participating institutions. A patient who had received collaborative treatment at one of these institutions was approached by the researcher. After agreeing to participate in the study, a face-to-face interview was conducted by trained and experienced researchers at each medical facility. Each interview was conducted after the end of the treatment and lasted approximately 15–20 min. Each participant was assigned a unique identification number, and the data were entered into an Excel sheet.

### 2.4. Statistical Analysis 

The baseline and clinical characteristics were summarized using descriptive statistics, including frequency and percentage. Hierarchical multiple regression analysis was used to explore the factors associated with overall patient satisfaction in collaborative KM-WM treatment. This model allowed for the adjustment of confounding variables, providing a clearer understanding of the relationships between predictors and patient satisfaction. The data were analyzed using SAS 9.4, with the significance level set at <0.05 and a 95% confidence interval (CI). 

### 2.5. Ethical Considerations 

The study was conducted across 15 medical institutions in Korea. Because each institution has its own institutional review board (IRB), it was mandatory to obtain approval from each participating institution. The study adhered to the principles outlined in the Declaration of Helsinki and received approval from the institutional review board of the following participating institutions: Pusan National University Korean Medicine Hospital (PNUKHIRB 2023-08-002-005); Kyung Hee University Korean Medicine Hospital (KOMCIRB 2023-080-030-001-HE001); Dongshin University Korean Medicine Hospital (NJ-IRB-23-06); Daegu Hanny University Hospital (DHUMC-D-23-010-PRO-01); Daejeon Jaseng Hospital of Korean Medicine (JASENG 2023-09-017-002); Mokpo Dongshin University Korean Oriental Hospital (DSMOH 23-03); Dong-Eui University Korean Medicine Hospital (DH-2023-05); Bucheon Jaseng Korean Medicine Hospital (JASENG 2023-09-016-002); Bundang Jaseng Hospital of Oriental Medicine (JASENG 2023-09-012-003); Wonkwang University Gwangju Medical Center (WKIRB 2023-10); Wonkwang University Jeonju Oriental Medicine Hospital (WUJKMH-IRB-2023-0007); Wonkwang University Korean Medicine Hospital (IRB 2023-10); Jaseng Hospital of Korean Medicine (JASENG 2023-09-015-001); Wonkwang University Jangheung Integrative Medical Hospital (exempted); and Haeundae Jaseng Hospital of Korean Medicine (JASENG 2023-09-014-001). Written informed consent was obtained from all participants. Privacy and confidentiality were ensured throughout the study, and each participant was fully informed about the study’s purpose.

## 3. Results

### 3.1. Baseline Demographic and Clinical Characteristics 

The baseline demographic and clinical characteristics of the participants are detailed in Table 1. Of the total 321 participants, the majority were female (60.06%) and aged 60 years or older (36.55%). Most participants had been visiting the medical institution for less than 6 months (68.13%). Regarding the treatment routes, 77.85% of the participants transitioned from KM to WM, whereas 22.15% transitioned from WM to KM (22.15%). The primary motivation for using collaborative treatment was access to a variety of treatment options (60%), followed by the desire for a comprehensive examination (29.38%). Additionally, a significant majority of the participants (89.72%) mentioned strong trust in KM.

### 3.2. Satisfaction with Collaborative KM-WM Treatment

Overall, patient satisfaction with collaborative KM-WM treatment was notably high (91.25%). Most participants were satisfied with the treatment effects, with 51.09% strongly agreeing and 42.68% agreeing that the care yielded significant outcomes. A substantial majority also reported that the time from hospital visits to treatment completion was appropriate (87.23%). Satisfaction with medical and administrative procedures was reported by 88.79% and 85.90%, respectively. Additionally, a greater proportion of participants (91.58%) felt that a pilot project for collaborative KM-WM treatment was necessary. However, satisfaction with hospitals’ promotion of collaborative KM-WM services was relatively lower, with only 34.69% expressing strong agreement (Table 2).

### 3.3. Need for Collaborative KM-WM Treatment 

A significant portion of the participants (90.66%) highlighted the need to include inpatient services in collaborative KM-WM treatment and care. The participants strongly emphasized the necessity (89.41%) of allowing KM-WM practitioners to determine the number of consultations based on patient needs. There was also a call for broader integration of the KM-WM pilot project, with 85.36% mentioning that the project should be expanded to different institutions. A greater proportion of participants expressed the need for the inclusion of primary care institutions (80.06%) and nursing hospitals (78.99%) in the KM-WM pilot project compared to dental care (56.07%). Furthermore, 89.41% of the participants reported the need to establish a “Collaborative Medical Institution Certification” to recognize institutions that effectively integrate collaborative KM-WM treatment (Table 3).

### 3.4. Factors Influencing Overall Patient Satisfaction

Table 4 presents the hierarchical multiple regression model for predicting the factors impacting the overall satisfaction with collaborative KM-WM. The delta R^2^ was significant in Model 2, with 64.0% of the variance in overall satisfaction (ΔR^2^ = 0.377). Trust in KM had a positive effect on patient satisfaction (*β*, 0.517; CI: 0.415–0.619) in Model 1. Satisfaction with treatment effects (*β*, 0.344; CI: 0.237–0.451), appropriate treatment time (*β*, 0.140; CI: 0.051–0.229), medical procedural efficiency (*β*, 0.227; CI: 0.126–0.328), and promotional activities (*β*, 0.175; CI: 0.101–0.250) significantly increased overall patient satisfaction with collaborative KM-WM treatment (each, *p* < 0.05). Conversely, no significant impact of administrative procedures on patient satisfaction with collaborative treatment was observed (*p* = 0.526). In addition, gender had a positive effect on overall satisfaction in Model 2, indicating that females have higher overall satisfaction than males (*p* = 0.048).

## 4. Discussion

We assessed patient satisfaction and needs regarding collaborative KM-WM treatment in the fourth phase of the pilot project. Overall, patient satisfaction with collaborative treatment was high and positively associated with treatment effects, treatment time, and medical procedural efficiency. In addition, a substantial proportion of patients indicated a strong need for collaborative treatment.

The first and second phases of the collaborative KM-WM pilot project reported satisfaction rates of 88.9% and 89.56%, respectively. However, this study revealed that patient satisfaction rates in the fourth phase were higher (91.25%) than those in the previous phases. A study conducted among healthcare providers in the first phase of the KM-WM pilot project revealed satisfaction with the treatment effect (59.3%), the need for expansion to other institutions (62.9%), and the inclusion of the primary care institutions (70.3%) [7,8]. The second phase reported satisfaction with the treatment effect (90.57%), the necessity of collaborative treatment (89.23%), expansion to other institutions (76.09%), and flexibility in determining the consultation of collaborative treatment (88.22%) [8]. Comparing the findings of this study with those from the first and second phases, there is a consistent trend of increased satisfaction and demand as the project progresses to the fourth phase. This consistency indicates that collaborative KM-WM care meets patient expectations, particularly concerning treatment effectiveness, quality and medical procedural efficiency. However, a study by Lee et al. reported slightly lower satisfaction levels with collaborative treatment for older adults with degenerative arthritis [17], whereas healthcare providers had differing perceptions of the necessity for collaborative treatment [20]. Furthermore, studies on collaborative KM-WM treatment among patients with disabilities, stroke, back pain, and nervous and digestive issues have shown varied levels of satisfaction, indicating the need for tailored approaches depending on the specific condition being treated [5,21,22,23]. This discrepancy may stem from differences in participants’ demographics and disease conditions. Older adults with chronic conditions might have expectations and experiences distinct from those of the more general population sampled in our study [21,24]. Further studies should assess and compare the effectiveness of collaborative treatment across diverse patient groups with different disease conditions. Aside from collaborative treatment, a study reported that the satisfaction rate with KM was 75% [25]. Similarly, a cross-sectional study involving 1000 Koreans revealed high levels of trust (69.3%) and positive perceptions of KM [26]. Compared with WM, remote treatment using only KM also demonstrated greater patient satisfaction, with the primary reason being thorough counseling and rigorous management [27]. Patients who received traditional medicine alongside conventional care had greater satisfaction than those who received only one type of treatment [28], which aligns with our findings. Therefore, collaborative treatment could offer better health outcomes in terms of both treatment effectiveness and satisfaction than treatment with only KM or WM. 

The public’s perception has shown high awareness of collaborative treatment practices but expressed concerns about the financial burden [15,21,24]. Despite financial concerns, the necessity and demand for collaborative treatment remain robust, with 91.58% of participants supporting the continuation of the collaborative KM-WM pilot project. Exploring the financial aspects of collaborative treatment, including patient willingness to pay and the cost-effectiveness of such treatments, is essential [8,14,22,24]. Owing to the limited number of studies, the economic factors affecting the need for collaborative treatment have not been thoroughly explored or evaluated [7,9]. Therefore, efforts are being made with a collaborative KM-WM pilot project to provide comprehensive evidence on the efficiency, safety, and cost-effectiveness of the treatment, including observational studies and clinical trials [4,16].

Our findings show that trust in KM was a prominent predictor of patient satisfaction, which aligns with previous literature that underscores the critical role of trust in healthcare settings [9,18,26]. For instance, Krot et al. suggested that trust in healthcare services can significantly increase patient satisfaction by fostering a positive therapeutic relationship and increasing engagement in treatment processes [29]. The positive associations between satisfaction and treatment effects, treatment time, procedural efficiency, and promotional activities support previous findings on the critical elements that drive patient satisfaction [9,18,30,31]. These findings indicate that effective treatment outcomes and efficient procedural practices are integral to patient satisfaction as they directly impact the quality and perceived value of treatment. In contrast, our results show a lack of significant influence of administrative procedures on patient satisfaction [9], likely because direct treatment experiences may take precedence over administrative concerns. In such collaborative care, patients might prioritize the effectiveness and efficiency of the treatment over administrative processes, which are often seen as peripheral to their primary healthcare experience [32]. Additionally, the positive impact of gender on patient satisfaction corroborates the findings of previous studies [31,33,34], which may be attributed to patient–provider interactions, holistic approaches, healthcare needs, and patient preferences. This observed effect warrants further exploration of gender-specific factors in collaborative treatment. 

Our findings have significant implications for health equity and policy in collaborative KM-WM treatment. The high satisfaction rates and positive demand for collaborative treatment highlight the need to expand these programs to more institutions and include outpatient services, as emphasized by 90.66% of the participants. This aligns with recommendations by the Health Insurance Review and Assessment Service for broader implementation to increase accessibility and continuity of care [4]. The demand for flexible consultation limits and institutional certification indicates a need to systematize and standardize collaborative care. Such measures could improve care quality and ensure that KM-WM practitioners can meet patient needs effectively [15,19]. The impact and benefits of collaborative treatment appear to vary, suggesting that outcomes may be influenced by patients’ needs and priorities, participating institutions’ strategies, and the particular conditions being treated [14,23,35]. In addition, with the operation of a monitoring center for KM-WM and evidence-based investigations, the KM-WM collaboration provides a systematic assessment of both clinical outcomes and economic dimensions, contributing valuable insights to the ongoing discourse on collaborative treatment. Nevertheless, the collaborative KM-WM pilot project presents an opportunity to enhance the overall healthcare system by combining the strengths of both medical approaches.

This study has several limitations. This study included patients from only fifteen institutions participating in the collaborative KM-WM pilot project, which may limit the representativeness of the sample and its generalizability to other regions, institutions, and the general patient population. The institutions were selected on the basis of their quality grade, collaborative treatment infrastructure, and research capabilities, potentially introducing selection bias. Patients who had difficulty reading and writing or who had mental or physical impairments were excluded from the study, which might omit the perspectives of a vulnerable patient population who could have different needs and levels of satisfaction with collaborative treatment [21]. These patients may represent a demographic with different health outcomes or unique barriers to accessing healthcare, potentially related to lower health literacy, socioeconomic status, or other contextual factors. By excluding them, this study may overlook important insights into how these challenges affect the health outcomes being investigated. Since this survey did not differentiate between various diseases treated, future research should address disease specificity and utilize more comprehensive evaluation tools to analyze its effectiveness. In addition, the timing of the data collection period could have introduced seasonal bias. The use of semi-structured tools and quantitative measures (such as Likert scales) may introduce response bias and limit the depth of understanding of patient experiences and perceptions. Therefore, further research should account for these aforementioned factors to provide a nuanced assessment of collaborative treatment.

In conclusion, the fourth phase of the collaborative KM-WM pilot project demonstrates high patient satisfaction and a strong demand for collaborative treatment and care. This study underscores that collaborative treatment effectively enhances patient perceptions and outcomes and supports the further expansion of the project to additional hospitals and primary care institutions. This study suggests delving further into disease-specific needs and patient demographics to tailor treatments more effectively and address any observed discrepancies in satisfaction across different conditions. Longitudinal, observational clinical studies and mixed-methods approaches with larger sample sizes could be more beneficial for providing a better understanding of patient experiences and effectiveness and for refining collaborative care strategies for optimal health outcomes. Additionally, exploring economic aspects, such as cost-effectiveness and patient willingness to pay, will also be crucial in developing sustainable and efficient models for collaborative treatment.

## Figures and Tables

**Table 1 healthcare-12-01901-t001:** Baseline demographic and clinical characteristics of the participants (N = 321).

Variables	Category	Frequency (%)
Gender	Male	127 (39.94)
	Female	191 (60.06)
Age (years)	20–29	19 (5.94)
	30–39	42 (13.13)
	40–49	62 (19.38)
	50–59	80 (25.00)
	≥60	117 (36.55)
Purpose of visit to medical institution	Treatment for an acute condition	171 (53.61)
	Treatment for a chronic condition	148 (46.39)
Duration of visit to medical institution (months)	1	112 (35.00)
	1–6	106 (33.13)
	6–12	37 (11.56)
	>12	65 (20.31)
Route of treatment	WM to KM	70 (22.15)
	KM to WM	246 (77.85)
The decision to receive collaborative KM-WM treatment	From healthcare professionals (doctors)	171 (53.61)
	Myself	148 (46.39)
Motivation for using collaborative KM-WM treatment	To access a variety of treatment options	192 (60.00)
	To obtain comprehensive and detailed care	94 (29.38)
	Recommendation from others	17 (5.31)
	Because of a pilot project institution	10 (3.12)
	Others	7 (2.19)
Trust in KM	Very high	127 (39.56)
	High	161 (50.16)
	Moderate	30 (9.35)
	Low	3 (0.93)
	Very low	0 (0.00)

KM, Korean medicine; WM, Western medicine; N may not be equal to the total due to missing values or responses.

**Table 2 healthcare-12-01901-t002:** Patient satisfaction with collaborative KM-WM treatment (N = 321).

Variables	Strongly Agree	Agree	Neutral	Disagree	Strongly Disagree
I’m satisfied with the treatment effects of the collaborative KM-WM care.	164 (51.09)	137 (42.68)	17 (5.30)	3 (0.93)	0 (0.00)
The time from the hospital visit to treatment completion was appropriate.	150 (46.73)	130 (40.50)	35 (10.90)	6 (1.87)	0 (0.00)
I’m satisfied with the medical procedures, such as examination and treatment.	159 (49.53)	126 (39.26)	28 (8.72)	7 (2.18)	1 (0.31)
I’m satisfied with the administrative procedures, such as registration and payment.	146 (45.77)	128 (40.13)	39 (12.23)	5 (1.56)	1 (0.31)
I’m satisfied with the promotion of collaborative KM-WM treatment by this hospital.	111 (34.69)	123 (38.44)	81 (25.31)	5 (1.56)	0 (0.00)
Overall, I’m satisfied with the collaborative KM-WM treatment based on the responses above.	149 (46.56)	143 (44.69)	25 (7.81)	2 (0.63)	1 (0.31)
I will recommend the collaborative KM-WM treatment to others.	166 (51.71)	128 (39.87)	17 (5.30)	10 (3.12)	0 (0.00)
I think a pilot project for collaborative KM-WM treatment is necessary.	173 (53.89)	121 (37.69)	21 (6.55)	5 (1.56)	1 (0.31)

Values denote frequency (percentage); KM, Korean medicine; WM, Western medicine; N may not be equal to the total due to missing values or responses.

**Table 3 healthcare-12-01901-t003:** Patient need for collaborative KM-WM treatment (N = 321).

Variables	Strongly Agree	Agree	Neutral	Disagree	Strongly Disagree
The future collaborative KM-WM pilot project should also include inpatient services.	155 (48.29)	136 (42.37)	23 (7.16)	6 (1.87)	1 (0.31)
KM-WM practitioners should be allowed to decide the number of consultations based on the condition, rather than the current biweekly limit.	138 (42.99)	149 (46.42)	27 (8.41)	7 (2.18)	0 (0.00)
The collaborative KM-WM pilot project should be integrated into different institutions.	139 (43.30)	135 (42.06)	35 (10.90)	11 (3.43)	1 (0.31)
Primary care institutions, such as general clinics and KM clinics, should be included in the collaborative KM-WM pilot project.	113 (35.20)	144 (44.86)	45 (14.03)	16 (4.98)	3 (0.93)
Nursing hospitals should be included in the collaborative KM-WM pilot project.	119 (37.30)	133 (41.69)	49 (15.36)	16 (5.02)	2 (0.63)
Dental care should be included in the collaborative KM-WM pilot project.	81 (25.23)	99 (30.84)	97 (30.22)	38 (11.84)	6 (1.87)
The government should establish a “Collaborative Medical Institution Certification” to recognize institutions with collaborative KM-WM treatment.	158 (49.22)	129 (40.19)	29 (9.03)	5 (1.56)	0 (0.00)

Values denote frequency (percentage); KM, Korean medicine; WM, Western medicine.

**Table 4 healthcare-12-01901-t004:** Hierarchical multiple regression model for predicting overall satisfaction with collaborative KM-WM.

Predictors	*β*, 95% CI	SE	*p*-Value
Model 1			
Age	−0.012 (−0.066–0.043)	0.028	0.676
Gender	0.114 (−0.023–0.250)	0.069	0.103
Duration of visit	−0.013 (−0.073–0.048)	0.031	0.685
Treatment route	0.012 (−0.152–0.177)	0.084	0.883
Trust in KM	0.517 (0.415–0.619)	0.052	<0.001 *
R^2^ = 0.263, *F* = 21.668, *p* < 0.001			
Model 2			
Age	0.003 (−0.035–0.041)	0.019	0.877
Gender	0.098 (0.010–0.194)	0.049	0.048 *
Duration of visit	−0.006 (−0.043–0.031)	0.022	0.981
Treatment route	−0.035 (−0.152–0.082)	0.059	0.554
Trust in KM	0.078 (−0.010–0.166)	0.045	0.076
Treatment effects	0.344 (0.237–0.451)	0.054	<0.001 *
Appropriate treatment time	0.140 (0.051–0.229)	0.045	0.002 *
Medical procedural efficiency	0.227 (0.126–0.328)	0.051	<0.001 *
Administrative procedures	−0.031 (−0.127–0.065)	0.049	0.526
Promotional activities	0.175 (0.101–0.250)	0.038	<0.001 *
R^2^ = 0.640, *F* = 53.249, ΔR^2^ = 0.377, *p* < 0.001			

*β*, coefficients; SE, standard error; CI, confidence interval; KM, Korean medicine; WM, Western medicine; * statistically significant at *p* < 0.05.

## Data Availability

The datasets generated during the current study are available from the corresponding author upon reasonable request.

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
