# Peer review of "Assessing Patient Satisfaction and the Need for Collaborative Treatment with Korean and Western Medicine"

_healthcare, 2024, doi:10.3390/healthcare12181901_

Round 1
Reviewer 1 Report
Comments and Suggestions for Authors
Thank you for this interested paper
Please structured your method in different sections: for example you need elaborate more for the data collection and ethical considerations. Explain the instrument little bit.
In the discussion, please link your significant finding with other literature and indicate if it has similar or different finding.
Comments on the Quality of English LanguageThe paper is well written
Author Response
"Please see the attachment."
We thank the reviewer for the thoughtful comments and suggestions. We have taken careful notice of the reviewers’ comments and have critically revised our manuscript. We believe that the manuscript has improved considerably.

Reviewer 2 Report
Comments and Suggestions for Authors
The research idea is needed at this stage to assess the performance of the system. Thank you for conducting the study and I have some comments and suggestions for the authors:
- The use of Korean medicine, or Korean traditional medicine, KM, western medicine, WM, KM-WM, medicine hospital, Korean traditional hospital, and so on was confusing for me. I believe all refer to the same two types of medicine here. This should be revised throughout the manuscript and introduce abbreviations (such as KTM and WM) then use them throughout the manuscript. Try to be consistent as much as you can to avoid confusion.
- You are not supposed to report the number of participants in the methods as this is part of your results.
- You methods section is very brief and much more details is needed to understand what you have done in the study.
How the participants were recruited for the study? What were their characteristics to be recruited and included or excluded?
How the questionnaire was developed, validated, piloted, and used?
How the data was collected from the patients? Who did the data collection? how long it took to collect the data? Where did the data were collected?
How the sample size was determined to be enough?
More details on the statistical analysis is needed to assess its suitability for the analysis? how did you deal with the response (did you d o any transformation?) what was the dependent variable in the regression?
Comments on the Quality of English Language
It is good but it can be improved.
Author Response

(The authors gave the same response as above.)

Reviewer 3 Report
Comments and Suggestions for Authors
Please see the attached file for my comments.

Author Response

(The authors gave the same response as above.)

Reviewer 4 Report
Comments and Suggestions for Authors
Dear Authors,
Your study exploring patient satisfaction and the necessity for collaborative treatment between Korean and Western medicine is both intriguing and significant, addressing crucial aspects of healthcare dynamics. I have several suggestions for your article quality improvement, as following:
Introduction Section: The introduction should specify the research gap this study intends to fill. Please provide a direct statement on how this study adds to the existing literature on KM and WM collaborative treatments. Please include more recent studies that focus on patient outcomes and satisfaction within collaborative healthcare systems.
Exclusion Criteria: Justify the exclusion of patients with reading and writing difficulties, discussing how this might affect the diversity and applicability of your findings - you can add this in to the Limitation Section
In Subsection 2.2. The description of the data collection process is brief and lacks specifics regarding how the integrity and consistency of the data were maintained across multiple institutions. Please elaborate on the training and standardization procedures for researchers who conducted the interviews and how inter-rater reliability was assured.
Conclusion Section: The current conclusion reiterates results without integrating them into broader clinical or research implications. Please revise to offer a synthesis of findings with practical applications and future research directions.
Good luck with your publication, kind regards
Author Response

(The authors gave the same response as above.)

Round 2
Reviewer 2 Report
Comments and Suggestions for Authors
The authors have properly addressed my comments.